# Detection of relativistic fermions in Weyl semimetal TaAs by magnetostriction measurements

T. Cichorek [1 ✉], Ł. Bochenek [1], J. Juraszek [1], Yu. V. Sharlai[2] & G. P. Mikitik [2 ✉]

Thus far, a detection of the Dirac or Weyl fermions in topological semimetals remains often elusive, since in these materials conventional charge carriers exist as well. Here, measuring a field-induced length change of the prototype Weyl semimetal TaAs at low temperatures, we find that its $c$-axis magnetostriction amounts to relatively large values whereas the $a$-axis magnetostriction exhibits strong variations with changing the orientation of the applied magnetic field. It is discovered that at magnetic fields above the ultra-quantum limit, the magnetostriction of TaAs contains a linear-in-field term, which, as we show, is a hallmark of the Weyl fermions in a material. Developing a theory for the magnetostriction of non-centrosymmetric topological semimetals and applying it to TaAs, we additionally find several parameters characterizing the interaction between the relativistic fermions and elastic degrees of freedom in this semimetal. Our study shows how dilatometry can be used to unveil Weyl fermions in candidate topological semimetals.

---

[1] Institute of Low Temperature and Structure Research, Polish Academy of Sciences, 50-422 Wrocław, Poland. [2] B. Verkin Institute for Low Temperature Physics and Engineering, Ukrainian Academy of Sciences, Kharkiv 61103, Ukraine. ✉email: t.cichorek@intibs.pl; mikitik@ilt.kharkov.ua

Until groundbreaking experiments regarding the two-dimensional material graphene, a study of relativistic quasiparticles has been limited to the high-energy physics[1]. This single-layer allotrope of carbon is a zero-gap semiconductor with a linear energy dispersion of conduction and valence bands connected one with the other at their extremities, and thus giving rise to the presence of low-energy quasiparticles governed by the relativistic Dirac equation[2]. Even more promising for quantum information processing are certain three-dimensional semimetals with nontrivial topology that host massless chiral fermions as quasiparticle excitations described by the relativistic Weyl equation[3,4]. Due to the breaking of either the inversion symmetry or the time reversal symmetry, a Weyl semimetal is characterized by the band-touching points known as Weyl nodes around which the nondegenerate-in-spin bands disperse linearly in all three momentum-space directions[5].

Because of the inherent chirality of Weyl quasiparticles and the emerged monopole-like structure of the Berry curvature, Weyl semimetals promise the wealth of novel phenomena. In particular, the surface Fermi arcs, that are directly observed using momentum-resolved photoemission spectroscopy, are recognized as a prime characteristic of this class of topological semimetals[5]. The negative longitudinal magnetoresistance[6–9] caused by the chiral anomaly, and unusual quantum oscillations produced by a cyclotron motion that weaves together the Fermi arcs and chiral bulk states[10,11] are other inherent properties of these materials. Recently, a unique type of acoustic collective mode called chiral zero sound has been theoretically proposed for Weyl semimetals with multiple pairs of Weyl nodes[12], and giant quantum oscillations of the thermal conductivity discovered in the prototypical Weyl semimetal TaAs have been explained with this chiral sound[13]. However, most of these experimental signatures of the Weyl electrons are often difficult to track down, in particular because the relativistic fermions coexist with conventional quasiparticles in topological semimetals. For example, although the negative longitudinal magnetoresistance in a parallel magnetic field was observed in a number of Weyl semimetals, its interpretation as the long-sought manifestation of the chiral anomaly remains controversial due to a possible inhomogeneous current flow in bulk crystals[14,15] as well as in view of alternative explanations of this effect[16]. At present, measurements of a quantum-oscillation phase are widely used to detect the Weyl fermions (see, e.g., references in review[17]) since this phase is noticeably affected by the Berry curvature. However, such experiments sometimes lead to ambiguous results. Therefore, and because of a rapidly growing number of candidate materials, new experimental methods to detect signatures of relativistic quasiparticles in topological semimetals are highly desirable. In this context, it was recently shown that the magnetization and magnetic torque measurements of Weyl semimetals upon entering the ultra-quantum limit state in high magnetic fields can be a useful probe for discerning the relativistic quasiparticles[18–20].

In this article, we draw attention to the magnetostriction, i.e., to the field-induced length change, which results from the interaction between the electron and elastic degrees of freedom in a crystal. Using TaAs as an example, we show that measuring this thermodynamic quantity, one can clearly distinguish between the relativistic and conventional electrons already in the field range where the Weyl fermions are confined at their zeroth Landau level, but the trivial quasiparticles are far below their ultra-quantum limit. Developing a theory of the magnetostriction for topological semimetals with a noncetrosymmetric crystal structure, we demonstrate how parameters characterizing not only Weyl electrons but also their interaction with elastic deformations

can be extracted from the magnetostriction measurements. A firm evidence for Weyl fermions is found with the measurements along the [001] direction where the largest length changes are observed. By contrast, the longitudinal expansion along the [100] direction is by an order of magnitude smaller in the highest field applied, but this $a$-axis magnetostriction experiences immense changes from large positive to large negative values with minute deviations of the applied magnetic field from the [001] direction. We suppose that the observed anisotropic magnetostrictive stress can be relevant for future high-field Weyltronic devices.

## Results

**Magnetostriction of nonmagnetic semimetals**. The magnetostriction of nonmagnetic conductive materials is directly related to changes in the density of charge carriers in a magnetic field. Specifically, a pocket $i$ of the Fermi surface makes the following contribution to the field-induced relative length change (Supplementary Note 1):

$$\frac{\Delta L}{L} = \Lambda_i \big( n_i(B) - n_i(0) \big), \tag{1}$$

where $B = \mu_0 H$ is the magnetic induction produced in the sample by the external magnetic field $H$, $n_i(B)$ is the $B$-dependent density of the charge carriers in this pocket, and the constant $\Lambda_i$ depends on the direction along which the magnetostriction is measured. Formula (1) results from a minimization of the energy consisting of the elastic energy proportional to $(\Delta L/L)^2$ and of the energy of the interaction between the elastic and electron degrees of freedom. This formula is written under the assumption that a deformation of the crystal shifts the appropriate electron band as a whole and does not change its shape. As a rule, this rigid-band approximation is quite accurate for real semimetals. Indeed, it was experimentally shown that the magnetostriction is very small if all charge carriers belong to a single band[21], and this small value characterizes the precision of the rigid-band approximation. [In this case, formula (1) predicts that the magnetostriction vanishes since $n_i(B) = n_i(0)$ due to the conservation of the carriers]. On the other hand, for a multiband material, this thermodynamic quantity is greatly enhanced[21] due to a band overlap and an electron redistribution between the bands at the switching-on of the magnetic field. Below we consider only such multiband materials since all the known Weyl semimetals contain several groups of the charge carriers.

**Distinctions between the Weyl and trivial electrons**. When several groups of electrons or holes exist in a conductive material, their Fermi energy $E_F$ (or the chemical potential $\zeta$ at nonzero temperature) generally depends on the magnetic field. At first, however, we will neglect this $B$ dependence of $E_F$ since as will be shown below, this simplified approach can provide a sufficiently accurate description of the magnetostriction. We start with a comparison of the magnetostrictions produced by the Weyl quasiparticles and by the trivial electrons for which the spectrum has the parabolic form (Supplementary Notes 2 and 3). For trivial electrons in high magnetic fields, their lowest Landau level rises above the Fermi energy if the parameter $\delta$ characterizing the electron magnetic moment $\mu_e = \delta(e\hbar/m_*)$ is less than 1/2 where $m_*$ is the cyclotron mass. This moment consists of its spin and orbital parts, the latter being due to the spin-orbit interaction. For such fields, the Fermi-surface pocket $i$ of the trivial electrons empties, and $n_i(B) = 0$ in this ultra-quantum limit. Thus, the magnetostriction of these electrons becomes constant,

$$\frac{\Delta L}{L} = -\Lambda_i n_i(0) \equiv a_i. \tag{2}$$

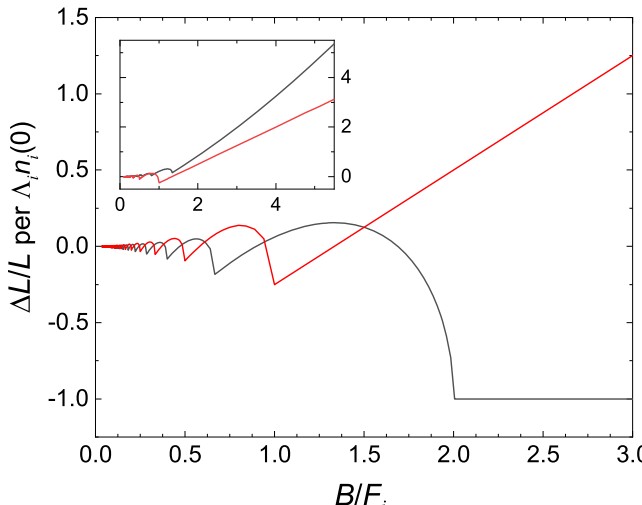

**Fig. 1 The magnetostriction of the Weyl electrons (red) and of the trivial electrons with parabolic spectrum (black) versus magnetic induction $B = \mu_0 H$ at zero temperature.** Here $F_i$ is defined by Eq. (3), the magnetostriction $\Delta L/L$ is calculated with equations of Supplementary Notes 2 and 3. For the trivial electrons, the parameter $\delta$ is assumed to be equal to zero. Note the relative phase shift of the oscillations shown by the red and black curves. Inset: The similar plot, but when $\delta = 0.75$ for the trivial electrons (black line). In this case $\Delta L/L = \Lambda_i n_i(0)(0.75 u^{-3/2} \sqrt{u + \delta - 0.5} - 1)$ for $B > F_i/(1.5 - \delta)$ where $u \equiv F_i/B$ (Supplementary Note 3).

This saturation of the magnetostriction takes place at $B > F_i/(0.5 - \delta)$, see Fig. 1, where

$$F_i = \frac{S_{\max,i}}{2\pi e \hbar} \qquad (3)$$

is the frequency of the quantum oscillations occurring at $B < F_i$, and $S_{\max,i}$ is the maximal cross-sectional area of the pocket $i$. On the other hand, if $\delta > 1/2$, the orbital $B$-dependent displacement $(e\hbar/2m_*)B$ of the lowest Landau level from the band edge $\varepsilon_0$ is less than the electron Zeeman energy $\mu_e B$, and at least one Landau level remains occupied by the charge carriers at any $B$. In this case,

$$\frac{\Delta L}{L} \approx a_i + \gamma_i B^{3/2}, \qquad (4)$$

for the magnetic fields $H \gg F_i/\mu_0$, with the factor $\gamma_i$ being dependent on $\delta$ (Fig. 1 and Supplementary Note 3).

In the weak magnetic fields $H \ll F_i/\mu_0$, if the quantum oscillations are suppressed by impurities or a temperature, the magnetostriction takes the form

$$\frac{\Delta L}{L} = -a_i \frac{3B^2}{8F_i^2}\left(\delta^2 - \frac{1}{12}\right) \equiv c_i B^2. \qquad (5)$$

If the oscillations are superimposed on a smooth background, this formula just describes this background.

In the case of the Weyl electrons, the zero Landau level coincides with the energy of the Weyl point for any $B$. Therefore, in the ultra-quantum regime when $\mu_0 H > F_i$, only this level is occupied by the electrons, their density $n_i(B)$ is proportional to $B$, and the magnetostriction is described by the formula

$$\frac{\Delta L}{L} = a_i\left(1 - \frac{3B}{4F_i}\right) \equiv a_i + b_i B, \qquad (6)$$

where $a_i = -\Lambda_i n_i(0)$, the parameter $F_i$ is still defined by formula (3) and coincides with the frequency of the quantum oscillations. A comparison of Eq. (6) with formulas (2) and (4) for the

parabolic spectrum shows that in the high-field region, the magnetostriction produced by Weyl fermions essentially differs from the magnetostriction of the trivial electrons (see also Supplementary Note 3).

For $H \ll F_i/\mu_0$, the magnetostriction of the Weyl electrons takes the form coinciding with Eq. (5) at the demarcative value of $\delta = 1/2$,

$$\frac{\Delta L}{L} = -\frac{a_i B^2}{16 F_i^2} \equiv c_i B^2. \qquad (7)$$

It is clear from formulas (2)–(7) that fits of the quadratic function $c_b B^2$ to the smooth background in the weak-field range and of a linear polynomial or a power function to the magnetostriction in the ultra-quantum regime allow one not only to detect the Weyl electrons but also to determine at least a part of the parameters $a_i$ and $F_i$ from the experimental data. With these parameters, the magnetostriction, including its subtle details as the quantum oscillations, can then be calculated in the entire range of the applied magnetic fields, using formulas of Supplementary Notes 1–3. A comparison of results of this calculation with the appropriate experimental data permits one to verify the existence of the conjectured Weyl electrons and to refine the values of the parameters for them.

**Advantages of the magnetostriction.** Consider a dependence of the magnetostriction on the volume of the Fermi-surface pockets (Supplementary Notes 2 and 3). In the case of the trivial electrons, one has $n_i(0) \propto |E_F - \varepsilon_0|^{3/2}$, $F_i \propto S_{\max,i} \propto |E_F - \varepsilon_0|$, and Eq. (5) yields for the weak magnetic fields,

$$\begin{aligned} c_i &\propto |E_F - \varepsilon_0|^{-1/2} \propto n_i^{-1/3}, \quad E_F > \varepsilon_0, \\ c_i &= 0, \qquad\qquad\qquad\qquad\quad E_F < \varepsilon_0, \end{aligned} \qquad (8)$$

where $\varepsilon_0$ is the edge of the energy band. (For the trivial holes the same formulas hold true but at the opposite relations between $E_F$ and $\varepsilon_0$). A similar increase of the coefficient $c_i$ with decreasing the charge-carrier density $n_i$ is obtained for the Weyl electrons from Eq. (7) since $n_i(0) \propto (E_F - \varepsilon_d)^3$, $F_i \propto S_{\max,i} \propto (E_F - \varepsilon_d)^2$ in this case, and hence

$$c_i \propto \pm |E_F - \varepsilon_d|^{-1} \propto n_i^{-1/3}, \qquad (9)$$

where $\varepsilon_d$ is the energy of the Weyl point, and the signs $\pm$ correspond to the electrons ($E_F > \varepsilon_d$) and holes ($E_F < \varepsilon_d$), respectively. The obvious difference between the Weyl and trivial charge carriers is that the function $c_i(E_F)$ changes its sign at the energy of the Weyl point $\varepsilon_d$, whereas this change does not occur for the trivial fermions. More importantly, however, the relation $c_i \propto n_i^{-1/3}$ reveals that the magnetostriction is substantially larger for small electron pockets than for large electron groups. That is why this quantity for elemental bismuth ($\Delta L/L \gtrsim 10^{-6}$ at 10 T[21,22]), the Fermi surface of which consists of small pockets, considerably exceeds the magnetostriction of metals ($\Delta L/L \sim 10^{-8}$ at 10 T[23]). This feature of the magnetostriction simply reflects the fact that the change of charge-carrier density in the weak magnetic fields (and at constant $E_F$) is less for a large Fermi-surface pocket than for a small one. In the ultra-quantum regime, when the change in the density becomes of the order of the density itself, the contribution to the magnetostriction generated by the large electron group can eventually exceeds the appropriate contribution of the small one, but such extreme fields are not currently available for dilatometric experiments.

Compare now the magnetostriction with those physical quantities that are proportional to the density of charge-carrier states at $B = 0$, $\nu_i = dn_i(E_F)/dE_F$ (e.g., with the non-oscillating part of the electrical conductivity[24]). This $\nu_i$ increases with $n_i$ both for

the trivial electrons ($\nu_i \propto n_i^{1/3}$) and for the Weyl quasiparticles ($\nu_i \propto n_i^{2/3}$). Hence, in measurements of those quantities, experimental signatures for small Fermi-surface pockets are extensively masked by the contribution of a large pocket when it exists in the material. Therefore, the above $n_i$ dependence of the magnetostriction makes it a useful tool for studying topological semimetals in which the Weyl points are in the vicinity of $E_F$.

In Supplementary Note 4 we also compare the magnetostriction with the magnetization $M$, the orbital part of which is not determined by the density of electron states as well, and which is considered as another thermodynamic probe of the Weyl electrons[18–20]. This comparison reveals the following distinctions between these quantities: i) Completely filled electron bands and, as clear from the above considerations, large Fermi pockets practically do not contribute to the magnetostriction. However, such pockets and even filled energy bands produce most of the magnetization, and this part of $M$ remains proportional to $B$ for the magnetic fields at which the Weyl electrons are in the ultra-quantum regime. Therefore, it is necessary to carry out a subtraction of extrapolated low-field magnetization from the high-field experimental data in order to extract the Weyl-electron contribution to $M$[17,25]. ii) If only small charge-carrier pockets exist in a semimetal, all these pockets can make large contributions to the magnetostriction. On the other hand, an attractive feature of the magnetization is that only its part produced by Weyl points is relatively large, whereas the part generated by small trivial-electron groups is insignificant. iii) Although the magnetostriction and the magnetization are similar in many respects, these quantities are associated with the different parts of the free energy of the conductive materials. The magnetization characterizes the electron energy in a magnetic field, whereas the magnetostriction results from the sum of the elastic energy and the energy of the interaction between the electron and elastic degrees of freedom in a crystal. For this reason, the detailed analysis of the magnetization and the magnetostriction can provide complementary information on the parameters of the Weyl points. In particular, the magnetostriction depends not only on the electron characteristics $F_i$, $n_i(0)$, but also on the constants $\Lambda_i$ which are determined by the elastic moduli of the crystal and by the constants of the deformation potential (Supplementary Note 1). These constants specify shifts of the energy bands under deformations in the conductive materials.

A measurement of the magnetic torque is the effective way of determining the transverse component of the magnetization[18–20]. However, this component and the magnetic torque vanish when the magnetic fields is aligned with a symmetry axis of a crystal. At magnetic fields tilted away from the symmetry axis, even equivalent pockets in a Weyl semimetal produce different contributions to the magnetization and to the magnetostriction, and a theoretical analysis of these quantities becomes complicated for the semimetals with multiple Weyl nodes. However, the magnetostriction as well as the longitudinal magnetization and the magnetotropic coefficient[26] remain nonzero at the magnetic field aligned with the symmetry axis, and such measurements of these quantities seem to be most convenient for an initial analysis of the Weyl fermions.

**Magnetostriction of TaAs along the [001] direction.** Let us exemplify the above considerations by an investigation of the field-induced length change of the Weyl semimetal TaAs. Like other transition-metal monopnictides NbP, NbAs, and TaP[27–30], tantalum arsenide crystallizes in a body-centered tetragonal structure that lacks a horizontal mirror plane and thus the inversion symmetry. This noncentrosymmetric structure is

essential to the existence of multiple pairs of the Weyl nodes divided into four pairs of the W1 points and eight pairs of the W2 points[3]. Among these materials, TaAs exhibits the largest separation of the Weyl nodes in momentum space, and the Fermi energy is sufficiently close to the Weyl points to produce a separate Fermi pocket encompassing each of these points. In addition to this, the cross-sectional areas of both the W1 (banana shaped) and W2 (nearly isotropic sphere) pockets are so small that the ultra-quantum limit for the Weyl electrons can be easily reached in experiments[15,31]. These electron pockets coexist with trivial hole pockets aligned along the nodal rings which would occur in TaAs if the spin-orbit interaction were absent[31]. When the magnetic field is parallel to the $c$ axis of TaAs, all the pockets in each of the W1 and W2 electron groups or in the group of the holes have equal densities $n_i(B)$ and extremal cross-sectional areas $S_{\mathrm{max},i}$. Let $F_{W1}$, $F_{W2}$, and $F_h$ denote the frequencies of quantum oscillations produced by the W1 and W2 electrons and by the holes, respectively. As discussed above, these $F_i$ also correspond to the magnetic inductions at which the W1 and W2 electrons and the holes enter the ultra-quantum regime. According to ref. [31], one has $F_{W1} \simeq 7\,\mathrm{T}$, $F_{W2} \simeq 5\,\mathrm{T}$, and $F_h \simeq 19\,\mathrm{T}$, with $F_{W2}$ being calculated but not measured for the [001] direction of the magnetic field. We emphasize that for $B\|c$, the entire field range up to 16 T, which is available in our experiments, may be considered as the low-field region for the holes.

Figure 2a shows the magnetostriction of TaAs measured along the [001] direction at 25 mK. The field-induced expansion is large and the relative length change $\Delta L/L$ amounts to about $5.5 \times 10^{-6}$ at $B = 16\,\mathrm{T}$. With the magnetic field aligned along the $c$ axis, the quantum oscillations reaching large amplitudes (~30% of the background signal at 3 T) are observable in the raw $\Delta L/L$ data (top panel) and even more sharply discernible in the derived coefficient $\lambda = \frac{1}{L}\frac{dL}{dB}$ (bottom panel) that represents the derivative of the charge-carrier density with respect to $B$. The sharp change observed in $\lambda$ at ~7.5 T and the linearly increasing signal at higher fields clearly show that the W1 and W2 electron groups enter the ultra-quantum regime, and the magnetostriction above 8 T is well approximated by the square polynomial $a + bB + c_h B^2$. The fit gives the values of $a$, $b$, and $c_h$ presented in Table 1. The presence of the linear-in-$B$ term in the magnetostriction clearly indicates the existence of the Weyl electrons, whereas the term $c_h B^2$ may be associated with the holes that are in the low-field range at $B \leq 16\,\mathrm{T}$. Thus, the presented experimental data do demonstrate the possibility of detecting the relativistic fermions with the magnetostriction.

Finally, we note that except for the oscillatory features, the overall $B$-dependence of the $c$-axis magnetostriction remains essentially unaltered up to the temperature 4.2 K [cf. Fig. 2b]. However, the log-log graph seems to reveal a deviation from the $B^2$ law expected in the low-field region (dashed black line). This finding might point to a very small value of $F_{W2}$ for $B\|$ [001]. However, the deviation of the $B$-dependence from the quadratic behavior is less than or of the order of $10^{-8}$, and hence a detailed insight into the low-field region requires larger TaAs crystals.

**Analysis of the magnetostriction for TaAs.** The theory of the magnetostriction presented in Supplementary Notes 1–3, permits us to describe quantitatively the experimental data for TaAs. We find that at $F_{W1} = 7.2\,\mathrm{T}$, the frequency of the calculated oscillations in the magnetostriction coincides with that observed experimentally. Since this frequency agrees with $F_{W1} = 7 \pm 0.5\,\mathrm{T}$ reported by Arnold et al.[31], we may be guided by that work in our analysis. As in all previous experiments with TaAs[13,19,31,32], the oscillations with the frequency $F_{W2}$ do not manifest themselves in our data on the magnetostriction, and we take $F_{W2} = 5\,\mathrm{T}$ to

maintain agreement with the results of the band-structure calculations[31]. Then, using formula (6) and the values of the

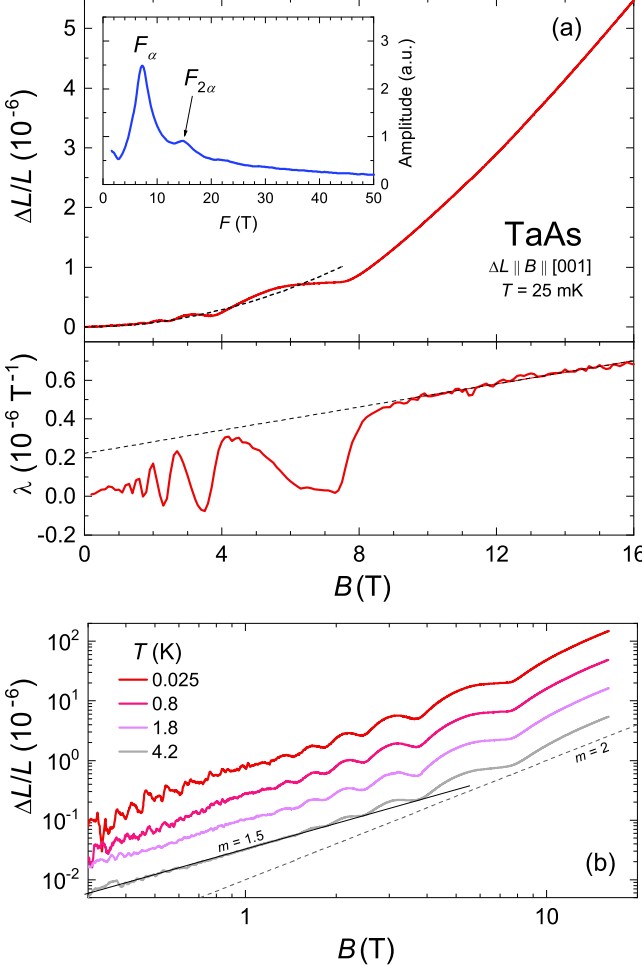

**Fig. 2 Magnetostriction in the Weyl semimetal TaAs. a** Top: Magnetic-field dependence of the relative length change $\Delta L/L$ of TaAs (sample 1) measured along the [001] direction at 25 mK in the parallel configuration ($B\|\Delta L$). Inset: FFT spectrum at $T = 25$ mK over a magnetic field range of 1–8 T. The oscillatory magnetostriction was obtained by subtracting the background $c_b B^2$ with $c_b = 1.79 \times 10^{-8}$ T$^{-2}$ (dashed line in the main panel) from the experimental data. Bottom: Corresponding magnetostriction coefficient $\lambda \equiv (1/L)dL/dB$. The straight dashed line approximating $\lambda(B)$ gives the intercept ($2.28 \times 10^{-7}$ T$^{-1}$) and has the slope ($2.95 \times 10^{-8}$ T$^{-2}$) which are close to the parameters $b$ and $2c_h$ in the quadratic polynomial $a + bB + c_h B^2$ (see the text and Table 1). **b** A log-log graph of the $c$-axis magnetostriction measured at different temperatures up to 4.2 K. Curves are offset for clarity. Note the slopes clearly distinct from the quadratic ($m = 2$) behavior expected for $B \ll F_i$. The dashed and solid black straight lines correspond to $\Delta L/L \propto B^m$.

constants $a$ and $b$ found above from the approximation of $\Delta L/L$ at $B > 8$ T, we arrive at the two linear equations in the parameters $a_{W1}$ and $a_{W2}$ characterizing the W1 and W2 electrons,

$$a_{W1} + a_{W2} = a,$$
$$-0.75\left(\frac{a_{W1}}{F_{W1}} + \frac{a_{W2}}{F_{W2}}\right) = b. \qquad (10)$$

These equations give $a_{W1} \approx -1.58 \times 10^{-6}$, $a_{W2} \approx -0.60 \times 10^{-6}$ (set 1 in Table 1). Note that at given $F_{W1}$ and $F_{W2}$, we have been able to determine all the unknown parameters for TaAs since there are only two nonequivalent groups of the Weyl electrons in this semimetal. Interestingly, the obtained $a_{W1}$ and $a_{W2}$ predict the value of the coefficient $c_b$ determining the low-field behavior $c_b B^2$ of the magnetostriction,

$$c_b = c_h - \frac{a_{W1}}{16 F_{W1}^2} - \frac{a_{W2}}{16 F_{W2}^2} \approx 1.83 \times 10^{-8} \text{ T}^{-2},$$

and this coefficient is close to that found experimentally (cf. the dashed line in Fig. 2a, top). With these parameters and with formulas of Supplementary Notes 1–3 at zero temperature, we calculate the magnetostriction of the Weyl electrons for all $B \leq 16$ T. To fit the magnitude of calculated oscillations to the experimental data, we use the dimensionless parameter specifying the scattering of the W1 electrons by impurities, $\gamma_{W1} = \pi T_{D,W1}/(E_F - \varepsilon_{d,W1})$, and find from the fit that $\gamma_{W1} = 0.025$ where $\varepsilon_{d,W1}$ is the energy of the Weyl points W1, and $T_{D,W1}$ is the Dingle temperatures for the W1 electrons. The similar ratio $\gamma_{W2} = \pi T_{D,W2}/(E_F - \varepsilon_{d,W2})$ for the W2 electrons is assumed to be equal to 0.1 in order to suppress the appropriate oscillations. Adding the hole contribution $c_h B^2$ to the calculated magnetostriction of the Weyl electrons, we find that the theoretical curve sufficiently well reproduces the experimental data in entire field range up to 16 T (compare the dashed green and solid red lines in Fig. 3) except for the 6–7.5 T interval where the last oscillation sets in (see the zoom in Fig. 3).

The above theoretical analysis of the magnetostriction is based on formulas obtained under the assumption of independence of the Fermi energy on the magnetic induction. This situation does can occur in a conductive material when it contains a large charge-carrier group that maintains the constancy of $E_F$. However, in TaAs all the electron and hole pockets are relatively small. In this case, a consideration must be given to the $B$ dependence of $E_F$ (i.e., of the chemical potential $\zeta$ if the temperature is nonzero) in analyzing the magnetostriction. This dependence $\zeta(B)$ can be found from the conservation condition of the total charge-carrier density,

$$\sum_i \left(n_i(\zeta, B) - n_i(\zeta_0, 0)\right) = 0, \qquad (11)$$

where $i$ runs all the electron and hole pockets, and $\zeta_0$ is the value of the chemical potential at $B = 0$. Since the dispersion law for the holes need not be well described by a simple parabolic dependence, we use the expression for $n_h(\zeta, B) - n_h(\zeta_0, 0)$ that

**Table 1 The values of the parameters for the calculation of the magnetostriction $\Delta L/L$ along the $c$ axis at $B\|c$ in neglect of the $B$ dependence of $\zeta$.**

| $a$ | $b$ | $c_h$ | $F_{W1}$ | set | $F_{W2}$ | $a_{W1}$ | $a_{W2}$ | $\gamma_{W1}$ | $\gamma_{W2}$ |
|---|---|---|---|---|---|---|---|---|---|
| $10^{-6}$ | $10^{-7}$T$^{-1}$ | $10^{-8}$T$^{-2}$ | T | # | T | $10^{-6}$ | $10^{-6}$ | | |
| −2.18 | 2.55 | 1.49 | 7.2 | 1 | 5 | −1.58 | −0.60 | 0.025 | 0.1 |
| | | | | 2 | 1.35 | −2.12 | −0.062 | | |

Values of $a_{W1}$ and $a_{W2}$ are found with Eqs. (10) from the coefficients of the polynomial $a + bB + c_h B^2$ that approximates the experimental data on the magnetostriction at $B > 8$ T.

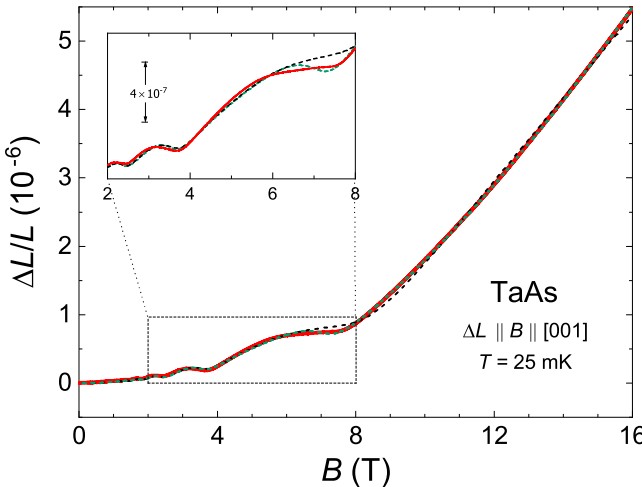

**Fig. 3 Comparison of the calculated magnetostriction for the case of $F_{W2} = 5$ T with the experimental data for TaAs.** The solid red line shows the c-axis magnetostriction $\Delta L/L$ of TaAs measured at $T = 25$ mK and $B\|c$. The dashed green line depicts this magnetostriction calculated at $T = 0$ with formulas of Supplementary Notes 1–3 (assuming the constancy of the chemical potential $\zeta$) for the values of the parameters presented in Table 1 (set 1). The dashed black line shows the magnetostriction calculated in Supplementary Note 7 for $T = 0$ and for the same values of $F_i$ and $\gamma_i$, taking into account the dependence $\zeta(B)$. The inset is a zoom into the last oscillations of the magnetostriction. Note the absence of a local minimum in the experimental curve above 6 T.

is valid at $B < F_h$ for any dispersion of these charge carriers,

$$n_h(\zeta, B) - n_h(\zeta_0, 0) = n_h(\zeta, B) - n_h(\zeta, 0) + n_h(\zeta, 0) - n_h(\zeta_0, 0)$$
$$\approx B^2 \left( \beta(\zeta_0) + \frac{d\beta(\zeta_0)}{d\zeta_0}(\zeta - \zeta_0) \right) + \nu_h(\zeta_0)(\zeta - \zeta_0),$$

where $\nu_h = \partial n_h(\zeta, 0)/\partial\zeta$ is the density of states for the holes in zero magnetic field, whereas the function $\beta(\zeta)$ defines the variation of the hole density in the low magnetic fields, $n_h(\zeta, B) - n_h(\zeta, 0) = \beta(\zeta)B^2$ [i.e., $\beta = c_h/\Lambda_h$ where $c_h$ has been introduced above].

Equation (11) and the general formula for the magnetostriction are explicitly written in Supplementary Note 5 for the case of $B\|c$, and with these expressions, one can calculate the magnetostriction in the entire range of the applied magnetic fields. As in the case of the simplified approach when $\zeta(B) = \zeta_0$, we set $F_{W1} = 7.2$ T, $F_{W2} = 5$ T, and take the same values of the constants $\gamma_{Wi}$. Apart from these parameters, the chemical potential $\zeta(B)$ depends also on the ratios $n_{W2}(\zeta_0, 0)/n_{W1}(\zeta_0, 0)$, $(\zeta_0 - \varepsilon_{d,W1})/(\zeta_0 - \varepsilon_{d,W2}) \equiv \nu$, and on the above-mentioned $\beta$, $d\beta/d\zeta$, $\nu_h$ normalized to $n_{W1}(\zeta_0, 0)$ where $\varepsilon_{d,W1}$ and $\varepsilon_{d,W2}$ are the energies of the Weyl points W1 and W2. Applying formulas of ref. [33] to the data of ref. [31], we may estimate a part of these parameters, viz., the density $n_{W1} \approx 2.5 \times 10^{18}$ cm$^{-3}$, the ratio $n_{W2}(\zeta_0, 0)/n_{W1}(\zeta_0, 0) \sim 0.15$, and the position of the chemical potential $\zeta_0$ relative to the energies $\varepsilon_{W1}$, $\varepsilon_{W2}$: $\zeta_0 - \varepsilon_{d,W1} \approx 28.4 \pm 3.5$ meV and $\zeta_0 - \varepsilon_{d,W2} \approx 11.9 \pm 1$ meV (Supplementary Note 6). These values of the parameters permit us to set $n_{W2}(\zeta_0, 0)/n_{W1}(\zeta_0, 0) = 0.15$ and $\nu = 2.5$ in our calculations of the magnetostriction. At these fixed ratios, the values of the other parameters are chosen so that the magnetostriction calculated at $T = 0$ matches the experimental data at $T = 25$ mK (Fig. 3 and Supplementary Note 7). Note that with the dependence $\zeta(B)$, the theoretical curve much better reproduces the plateau above about 6 T than in the case of the constant $\zeta$. The derived dependence of the chemical potential

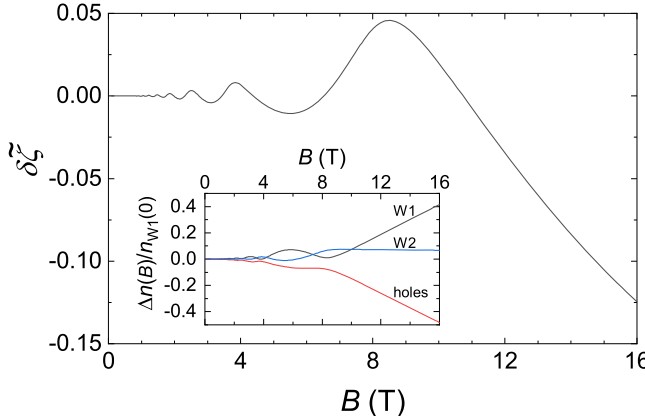

**Fig. 4 The field-induced shift of the chemical potential relative to its value $\zeta_0 \equiv \zeta(0)$ at zero magnetic field in TaAs.** Shown is the normalized shift $\delta\tilde{\zeta} \equiv (\zeta(B) - \zeta_0)/(\zeta_0 - \varepsilon_{W1})$ that is calculated together with the dashed black line in Fig. 3. Inset: Dependences of $(n_{W1}(B) - n_{W1}(0))/n_{W1}(0)$, $(n_{W2}(B) - n_{W2}(0))/n_{W1}(0)$, and $(n_h(B) - n_h(0))/n_{W1}(0)$ on B. Note that although $\delta\tilde{\zeta}$ decreases with increasing field above 8.5 T, $n_{w1}(B)$ simultaneously increases due to the growing capacity of the Landau levels. According to the definition (Supplementary Note 2), $n_h(0) < 0$.

on the applied magnetic field is presented in Fig. 4. It is seen that due to condition (11), the largest electron group W1 induces the oscillation with the same frequency 7.2 T for the other charge carriers. The analysis of the obtained parameters in Supplementary Note 7 enables us to find the constants $\Lambda_i \equiv \Lambda_i^c$, which determine contributions of the W1 and W2 electrons and of the holes to the c-axis magnetostriction of TaAs (Table 2). This analysis also reveals that the obtained values of $\beta$, $d\beta/d\zeta$, and $\nu_h$ can be understood from simple estimates, assuming the simplest parabolic dispersion of the holes in TaAs. We also obtain a good fit of the magnetostriction calculated at a finite dimensionless temperature $t = T/(\zeta_0 - \varepsilon_{W1}) = 0.015$ to the magnetostriction measured at the temperature $T = 4.2$ K (Supplementary Fig. 8) and therefore find the independent estimate of $\zeta_0 - \varepsilon_{W1} \approx 24$ meV, which is only a little less than the value $28.4 \pm 3.5$ meV derived above. With this $\zeta_0 - \varepsilon_{W1}$ and $\gamma_{W1} = 0.025$, we arrive at the Dingle temperature $T_{D,W1} = (\zeta_0 - \varepsilon_{W1})\gamma_{W1}/\pi \approx 2.2$ K, the value of which is comparable with $T_{D,W1} \approx 3.2$ K obtained for the W1 electrons in ref. [31]. (As to $T_{D,W2}$, we tentatively find $T_{D,W2} \sim 3.6$ K; Supplementary Note 7.)

Interestingly, if one decreases the value of the parameter $F_{W2}$, even better fits of the calculated magnetostriction to the experimental data can be obtained, and the best fit is reached at $F_{W2} \approx 1.35$ T (the dashed black line in Fig. 5). Thus, we conclude that the two sets of the parameters are worth considering (Table 1). The first set ($F_{W2} = 5$ T) is completely consistent with the Fermi-surface calculations of ref. [31], whereas the second set with $F_{W2} = 1.35$ T provides the best fit of the theoretical curve to our experimental data on the magnetostriction. The appropriate $\Lambda_i^c$ for the second set are presented in Table 2, and in contrast to set 1, the analysis of $\beta$, $d\beta/d\zeta$, and $\nu_h$ obtained for set 2 reveals that the dispersion of the holes should essentially deviate from the parabolic law. However, apart from the problem of choosing the value of $F_{W2}$, a comparison of all the data presented in Figs. 3 and 5 lead to the conclusion that in the first approximation, one can neglect the dependence of the chemical potential on the magnetic field (see also Supplementary Note 7). In other words, the simplified approach, within which $\zeta$ is independent of $B$, is sufficiently well justified for describing the magnetostriction.

**Table 2 The values of $\Lambda_i^c$ and $\Lambda_i^\perp$ obtained from the c-axis and a-axis magnetostrictions, respectively.**

| set # | $F_{W2}$ T | $10^{24}\Lambda_{W1}^c$ cm³ | $10^{24}\Lambda_{W2}^c$ cm³ | $10^{24}\Lambda_h^c$ cm³ | $10^{24}\Lambda_{W1}^\perp$ cm³ | $10^{24}\Lambda_{W2}^\perp$ cm³ | $10^{24}\Lambda_h^\perp$ cm³ |
|---|---|---|---|---|---|---|---|
| 1 | 5 | 0.71 | 1.47 | −3.7 | | | |
| 2 | 1.35 | 0.89 | 1.42 | −1.0 | −2.1 | 11.2 | 4.2 |

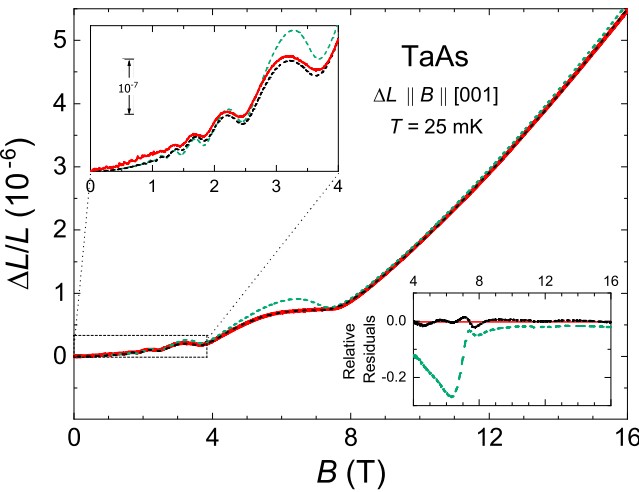

**Fig. 5 Comparison of the calculated magnetostriction for the case of $F_{W2} = 1.35$ T with the experimental data for TaAs.** The solid red line shows the c-axis magnetostriction $\Delta L/L$ of TaAs measured at $T = 25$ mK and $B\|c$. The dashed green and black lines have the same meaning as in Fig. 3, but they are calculated for set 2 in Table 1. The upper inset is a zoom into the low-field region, and the lower inset compares the appropriate relative residuals, showing a very good agreement in the entire field range when $\zeta(B)$ is considered (dashed black line).

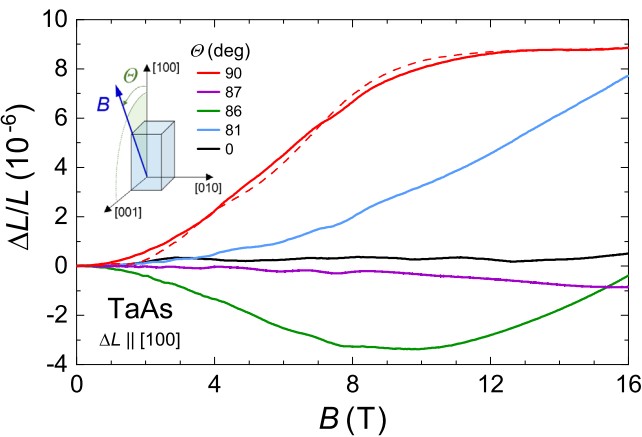

**Fig. 6 Angle-dependent magnetostriction of TaAs measured along the a axis.** Magnetic-field dependence of the relative length change $\Delta L/L$ of TaAs (sample 2) is measured along the [100] direction at 25 mK for various angles $\Theta$ between the direction of $B$ and the $a$ axis. The magnetic field lies in the plane (010). Note the exceptional changes of $\Delta L/L$ over the small-angle range near $\Theta = 90°$. (The corresponding results for $\Theta \le 45°$ are shown in Supplementary Fig. 3). The dashed red line shows the $a$-axis magnetostriction that is calculated at $T = 0$ for the field aligned with the $c$ direction, using $F_{W1} = 7.2$ T, $F_{W1} = 1.35$ T, $\gamma_{W1} = 0.1$ and $\gamma_{W2} = 0.2$ (Supplementary Note 8).

In numerous experiments (see review[17] and references therein), the phase of quantum oscillations in topological semimetals was measured to distinguish between the Weyl (Dirac) fermions and trivial quasiparticles. Such investigations are based on the fact that in the case of the Weyl fermions, the nonzero Berry phase of the electron orbits in a magnetic field leads to a shift of the phase of the oscillations by $\pi$ as compared to the phase corresponding to the trivial electrons[17,34]. Obviously, in agreement with formulas of Supplementary Notes 2 and 3, this nontrivial phase shift should also occur for the oscillations in the magnetostriction. For the case of TaAs ($\Delta L\|B\|c$), the insets in Figs. 3 and 5 clearly demonstrate that the phase of the oscillations calculated with formulas for the Weyl electrons really coincides with the experimental one. We emphasize that if the nontrivial phase shift were absent in the measured oscillatory magnetostrictions, the theoretical and experimental curves would be mutually displaced in phase like the red and black lines in Fig. 1. Therefore, the coincidence of the phases, among other manifestations of the Weyl electrons in our magnetostriction measurements, also proves their existence in TaAs.

## Discussion

Above we found the two sets of the parameters for which the magnetostriction measured along the c axis can be well described theoretically. To choose between these two set, we have measured $\Delta L/L$ along the a axis with the magnetic field still aligned with the [001] direction. In this case, only the values of $a_{W1}$, $a_{W2}$, and $c_h$ can change due to a change of the constants $\Lambda_{W1}$, $\Lambda_{W2}$, and $\Lambda_h$. All the other parameters determining this relative length change should remain the same as in the case of the c-axis

magnetostriction. Note that the knowledge of the constants $\Lambda_i$ for the magnetostriction measured not only along the c axis but also along the a direction makes it possible to find the constants of the deformation potential that describe effects of strains on the band structure of TaAs (Supplementary Notes 1 and 8).

Figure 6 shows the magnetostriction measured along the a axis at $T = 25$ mK. We note that a field-induced length change is small ($\sim 0.5 \times 10^{-6}$ at $B = 16$ T) in magnetic fields applied along the dilatation direction (black curve). In addition, there is a complex $B$ dependence of $\Delta L/L$ in the entire field range. However, when the dilatometer is rotated by the angle $\Theta = 90°$ to the desired sample orientation $B\|c$, the a-axis magnetostriction exhibits a substantial enhancement with a behavior close to the $B^2$ law between 0.5 and 5 T. This behavior is followed by a tendency to the saturation at about $9 \times 10^{-6}$ above 12 T (red curve). Another remarkable feature of the a-axis magnetostriction is its high sensitivity to small deviations of the applied field from the [001] direction. Such deviations cause immense changes in the magnetostriction from large positive to large negative values. For example, the violet curve at $\Theta = 87°$ illustrates the field-induced contraction of TaAs that is about $-1.0 \times 10^{-6}$ at $B = 16$ T. Moreover, a drastic change of the negative magnetostriction occurs when the magnetic field is just marginally tilted further from the [001] direction ($\Theta = 86°$, green curve). At $\Theta = 81°$ (blue curve), we again observe a large expansion of TaAs. In contrast with this a-axis magnetostriction, the $B$-induced length changes along the c axis do not exhibit any sensitivity to small deviations of $B$ from the c axis. In Fig. 7, we present the angle-dependent magnetostriction of TaAs measured along the [001] direction at 4.2 K. It is seen that there are no qualitative changes in this

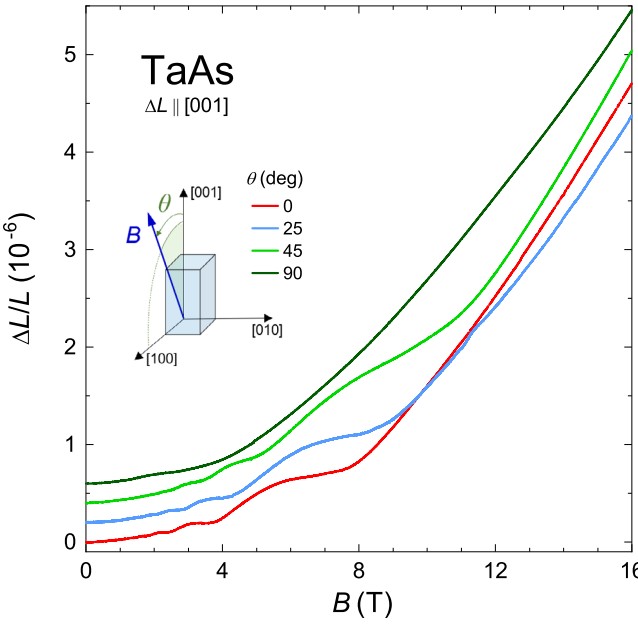

**Fig. 7 Angle-dependent magnetostriction of TaAs measured along the *c* axis.** Field-induced relative length change of TaAs (sample 3) along the [001] direction is measured at $T = 4.2\,K$ and at various angles $\theta$ between the direction of $B$ and the *c* axis. The magnetic field lies in the plane (010). For clarity, the different $\Delta L/L$ curves at $\theta > 0$ were shifted subsequently by $0.2 \times 10^{-6}$.

magnetostriction even at large deviation angles. (Now the tilt angle $\theta$ of $B$ is measured from the *c* axis.)

Varying only the constants $\Lambda_i$, we have calculated the *a*-axis magnetostriction at $B\|c$ for both sets of the parameters with $F_{W2} = 5\,T$ and $1.35\,T$ (Supplementary Note 8). A modification of the values of $\gamma_{W1}$ and $\gamma_{W2}$ is also admitted since Figs. 2 and 6 show the magnetostrictions for the different samples. For the set with $F_{W2} = 5\,T$, we have not been able to match well the theoretical curve with the experimental data (cf. Supplementary Fig. 9). For the second set, the theoretical curve can approximately reproduce these data (the dashed red line in Fig. 6) for certain values of $\Lambda_i \equiv \Lambda_i^{\perp}$ (Table 2) and for the increased $\gamma_i$ ($\gamma_{W1} = 0.1$ and $\gamma_{W2} = 0.2$). Thus, the obtained results seem to argue in favor of set 2 with $F_{W2} = 1.35\,T$. However, due to the extreme sensitivity of the *a*-axis magnetostriction to the field orientation relative to the [001] direction, its true $B$ dependence at $B\|[001]$ may essentially differ from the experimental curve ($\Theta = 90°$) presented in Fig. 6. Thus, a more elaborate approach to achieve a perfect magnetic-field orientation is required in order to reliably exclude the possibility of set 1. In Supplementary Note 8, we discuss a possible cause of the unusual high sensitivity of the *a*-axis magnetostriction to a small tilting of $B$ about the *c* axis.

The field-induced shift of the chemical potential presented in Fig. 4 clearly demonstrates the essential redistribution of the charge carriers between the bands of the holes and the W1 electrons. Although we found above that this effect can be of minor significance for the magnetostriction of TaAs, it can be relevant to understanding some other field-dependent properties of Weyl-semimetal-candidate materials. In particular, the charge-carrier redistribution leads to a nonzero longitudinal magnetoresistance of a semimetal if the quasiparticles in its different bands have dissimilar mobilities, and this magnetoresistance can be negative even for trivial charge carriers. Indeed, if with increasing $B$, the trivial electrons of a lower mobility are transferred to another band with higher mobility, the longitudinal conductivity of this material increases (see, e.g., Supplementary

Fig. S10 in Supplemental Material to ref. [35]). Therefore, the redistribution should be considered before giving any arguments in favor of the chiral anomaly in strong magnetic fields. In the case of TaAs, our study reveals that for $B\|c$, i.e., the most promising configuration for the chiral anomaly[15], both the absolute value $|n_h(B)|$ of the negative hole density and the density of the W1 electrons increase in high magnetic fields (Fig. 4, inset). In this situation, one may expect that the longitudinal conductivity along the [001] axis will increase above about 8 T for any relation between mobilities of the electrons and holes. Interestingly, the negative longitudinal magnetoresistance for $B\|c$ was really observed in TaAs at $7.5\,T < B < 25\,T$ in careful measurements that used the focused-ion-beam lithography to eliminate experimental artifacts due to electrical current inhomogeneities[15].

In summary, our study of the magnetostriction along the main crystallographic directions of TaAs shows that this quantity can be an effective probe of the massless quasiparticles in Weyl semimetals, if the Weyl points lie near the Fermi level. This statement holds even though conventional charge carriers exist in a semimetal. In this situation, even in moderate magnetic fields, which are too weak to confine large groups of massive quasiparticles at their zeroth Landau levels, the magnetostriction contains a linear-in-field term that identifies the presence of relativistic fermions. Moreover, in this case, the experimental magnetic-field and temperature dependences of the magnetostriction, including their subtle details, can be reproduced theoretically. Comprehensive dilatometric investigations of topological semimetals also shed light on dependences of the Weyl points on an applied stress and hence predict how the appropriate quantum-oscillation frequencies will change under a uniform compression of the material. It is also worth noting that our theory is applicable to Dirac semimetals. Therefore, in a broader perspective, detection of relativistic fermions in candidate topological materials with the magnetostriction can set the stage for their further investigations, including electronic applications.

## Methods

**Crystal synthesis**. TaAs single crystals were synthesized with the chemical vapor transport method following the procedure described elsewhere[36–38]. The chemical composition of the crystals was examined by electron-probe microanalysis with energy-dispersive x-ray spectroscopy. The ratio of 1:0.99 between Ta and As was found, indicating the correct stoichiometric chemical composition. The body-centered tetragonal structure (space group $I4_1md$, No. 109) of TaAs single crystals was confirmed by room-temperature x-ray diffraction. No other phases were detected, and the lattice parameters $a = 3.4348\,Å$ and $c = 11.6412\,Å$ are in good agreement with the literature values[36–38]. The crystal orientation was determined by Laue diffraction (cf. Supplementary Fig. 1, right).

**Sample characterization**. All samples used in this study were obtained from the same growth batch. The residual-resistivity ratio of typical sample RRR $= \rho_{300\,K}/\rho_{2\,K} \simeq 12$ was determined from a zero-field resistance measurement along the *c* axis. The transverse magnetoresistance MR of about 39 600% was measured at 9 T and 2 K (Supplementary Fig. 2). These parameters are in good agreement with the published literature values, and hence point at good quality of our TaAs single crystals.

**Magnetostriction measurements**. The angle-dependent field-induced length change was measured with a commercial capacitance dilatometer which enables a length resolution of 0.02 Å[39]. The magnetostriction of the three rectangular TaAs samples having a length of ~1.7 mm was studied along the [100] and [001] directions. We performed the magnetostriction measurements down to 25 mK in a dilution refrigerator (Kelvinox 400 HA, Oxford Instruments) inserted into a superconducting magnet for fields up to 16 T. A field-sweep rate as small as $0.5\,mTs^{-1}$ was used, and the highest temperature of our experiments was 4.2 K.

The capacitive dilatometer cell is compact enough to be mounted on an attocube rotator, and thus enables the study of the field-induced length changes as a function of the tilted angle. However, since TaAs single crystals with a typical cross-section of about $0.7 \times 0.9\,mm^2$ were mounted between two capacitor plates with the diameter of 20 mm, the samples cannot be oriented perfectly. The limited accuracy of orientation of the sample surfaces with respect to the dilatometer setup can cause the maximum error of 5°.

**Numerical calculations**. The $B$-dependences of the magnetostriction for TaAs have been numerically calculated, using our own code elaborated with formulas of Supplementary Notes 1–3, 5 and results of refs. [40–42].

## Data availability

The data that support the plots within the main manuscript or the supplement and other findings of this study are available from the corresponding authors upon reasonable request.

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

## Acknowledgements

This work was supported by the Polish National Science Centre (Project No. 2016/21/B/ST3/02361). Y.V.S. is highly grateful for the support from the Polish Academy of Sciences and the Safe Passage Fund of the U.S. National Academy of Sciences.

## Author contributions

T.C. conceived the project. L.B. performed the magnetostriction experiment with help from T.C. and J.J.; J.J. grew the single-crystal samples and did the electrical transport measurements. G.P.M. and Y.V.S. worked out the theory for the magnetostriction of noncentrosymmetric topological semimetals, and Y.V.S. did all numerical calculations for TaAs. T.C. and G.P.M. wrote the manuscript with an input from all the coauthors.

## Competing interests

The authors declare no competing interests.
