## [Peer Review File · Nature Communications]

Detection of relativistic fermions in Weyl semimetal TaAs by magnetostriction measurementsREVIEWER COMMENTS

Reviewer #1 (Remarks to the Author):

Cichorek et al. measured magnetostriction of a prototype Weyl semimetal TaAs in this manuscript. They demonstrated that the behavior of the magnetostriction in the quantum limit where only a single Landau level is occupied by carriers could be used to determine the relativistic fermions in it. A theory based on the rigid band approximation is set up to fit their experimental data, especially the magnetostriction measured along the [001] direction. The measurement and the corresponding theory are interesting, however, the following comments should be responded properly before its publication.

1, The authors argue that the lowest Landau level may rise above the Fermi energy when the δ is less than $1/2$ for trivial electrons. Do you mean the Fermi energy is fixed in the system? It is not physical since there are electrons without the magnetic field if all the electrons belong to a single band. Where do these electrons go in this extremely large field? Actually, you argue below Eq. (1) "formula (1) predicts that the magnetostriction vanishes since $n_i(B) = n_i(0)$ due to the conservation of the carriers". The discussion for $\delta < 1/2$ is inconsistent with this argument.

2, If $\delta > 1/2$, at least one level remains occupied at fixed Fermi energy case, and in the ultra-quantum limit, the magnetostriction is proportional to $B^{3/2}$. Further, it is linearly dependent on the magnetic field for Weyl electrons. However, it is hard to distinguish from each other. We can see from the inset of Fig. 1 that there is no obvious difference between the Weyl and trivial electrons just from two line shapes in the quantum limit. Hence, I think if you fit data of the magnetostriction along [001] direction by using $B^{3/2}$ but not B , you may obtain almost the same value of the coefficient b . I know if you fit the data with the trivial model, the mismatch of the phase in the oscillation part before the quantum limit will occur, which could exclude the trivial one. Then one could find out the Weyl electrons directly from the phase shift of the oscillation and it seems there is no need to determine the relativistic electrons from the data fit in the quantum limit.

3, I find the authors also plotted the derived coefficient λ versus the magnetic field in the bottom of Fig. 2(a). The behavior of λ in the quantum limit may help one to distinguish the Weyl electron from the trivial one. But I am not sure whether it is true. Further, it is linear in the quantum limit. Does the intercept of the straight line on the vertical coordinate is the coefficient b ? If the answer is true it should be compared with the one from the fit of magnetostriction above 8T?

4, The authors mentioned that for magnetization not the magnetostriction, a large electron group (or even a filled energy band) will give a large contribution to the magnetization proportional to B for Weyl electrons in the ultra-quantum regime, which may be indistinguishable from the term $B \ln(B)$ produced by the Weyl electrons. However, both the magnetostriction and the magnetization are thermodynamic probe. Why does the large electron group not affect the magnetostriction?

5, I find the authors emphasized that the magnetostriction relates to n_i , the density of the carriers, but not the density of the states like the resistivity. What is the advantage of it?

6, There are two kinds of Weyl nodes with different cross-sectional areas contributing to the magnetostriction. Especially, the shape of the W1 pocket is banana-like. How does this shape affect your results?

7, The authors fit the measured magnetostriction along [001] in Fig. 3 by a constant chemical potential or Fermi energy. Further, they argued that the B -dependence of the chemical potential could be neglected. It could be clearer if they also plot the calculated magnetostriction for the fixed electron density in Fig. 3. Actually, the Fermi energy alters a lot in this ultra-quantum limit shown in Fig. 4. Hence, the direct comparison may be more compelling.

8, The magnetostriction measured along the a axis show strong dependence on the angle between the magnetic field and the a axis. Particularly, it is positive at 81 degree, negative at 86 degree, and

positive again at 90 degree. Why does the sign of the magnetostriction rely on this angle so strongly? Further, this magnetostriction measured along the a axis could not be fitted by just varying the cross sectional area. A proper fit could be realized by using a completely different set of parameters. It is not convincing.

Reviewer #2 (Remarks to the Author):

In their manuscript “Unravelling relativistic fermions in Weyl semimetal TaAs by magnetostriction measurements”, T. Cichorek and coauthors investigate topological semi-metals via magnetostriction experiments in the quantum limit. The magnetostriction is a direct and directional probe of the free energy, and hence is sensitive to the details of the electronic dispersion. In particular, there is a free energy anomaly in the ultra-quantum limit that distinguishes it strongly from that of topologically trivial electrons. The paper works out the theory and applies it to TaAs, concluding that there are indeed Weyl orbits.

While I personally like the paper and subject very much, I unfortunately do not see this as a sufficient increment over the status-quo of the field to warrant a publication in Nature Communications. It definitely deserves publication, yet probably a more specialized journal is more appropriate.

Let me lay out my rationale. The anomalous behavior of Landau levels formed from Weyl fermions is well established, also by the authors works, and has been observed theoretically in the magnetization and the magnetic torque. There, the same mechanism leads to an anomaly that involves a sign change and very sharp features as the sample enters the quantum limit. The novelty of this work lies in its observation in magnetostriction, yet this is merely choosing a different channel to probe the free energy anomaly. Certainly the paper does not show any new physics about TaAs or ultra-quantum-limit physics, it shows that the magnetostriction is compatible with published results. This is wonderful, yet by itself too much of a nuance for a Nature Communication paper.

Before going into the details, I want to emphasize a main point of critique. I find the description of previous works, including some of the authors, to be brushed away too rigorously in the introduction. It is stated that magnetic susceptibility is a useful probe, yet then these signatures are supposedly “subtle and indirect”. Accidentally convolving the magnetic anomalies with the issues of negative LMR here is dishonest and does not do the paper service. The later discussion on the comparison between forms of magnetization and magnetostriction appears very much as a “sales argument” rather than an open discussion. The goal of this comparison is to argue that magnetostriction is superior as a technique, as is already evident from the title. I disagree with this spirit, both are interesting probes in their own right that elucidate different aspects of a physical object. As the physics is identical, the emphasis is on practical aspects such as field alignment and background subtraction. The authors are experts in modelling of magnetization and could have with the same ease explained the longitudinal magnetization or likely even the more complex torque. From a practical point of view, magnetization measurements are much simpler and prevalent compared to magnetostriction. Even so, the authors conclude that the fitting parameters do not yield a self-consistent picture once the magnetostriction along the a-direction is taken into account.

I would suggest to rewrite this paper to highlight what new physics we can learn. Magnetostriction does not need to be “superior” to magnetization; maybe combining the techniques truly unlocks new insights.

- The rigid band approximation is well justified in metals, yet I wonder if it is as simple in the case of Weyl fermions. For once, the involved energy scales are meV, on the order of 10^{-3} of the total bandwidth. Is it obvious that on these energy scales rigid band is always justified? These nodes may also be moved along equal energy contours. It may be, but a bit of justification would be helpful.
- The authors stress about eq (6) that it allows to obtain the FS cross-sections F_i , which are usually measured by quantum oscillation frequencies, even when there are no oscillations. This appears a bit misleading. It essentially calculates the area from the knowledge of the carrier density and computes the filling required to obtain it. Similarly, eq (5) would in principle allow to obtain the F_i without oscillations, yet it similarly needs stringent assumptions on the dispersion. Lastly, these are band-

dependent contributions which are not directly measurable by experiment, which always gets the sum. This sentence goes too far in my opinion and should be removed.

- I find it strange that W_1 is observed so vividly yet W_2 is completely absent. This might be due to that difference in scattering rates as the authors suggest (yet the a-direction magnetorestriction yields γ values that are incompatible with this and would suggest stronger oscillations of W_2). It would however require strongly k-selective scattering mechanisms.

- Conveniently, there are 2 Weyl pockets which allows to solve the polynomial for a_{W1} and a_{W2} . While implicitly stated in the discussions about field angles, it would be beneficial to state this clearly when discussing the calculation.

- I highly appreciate the robust treatment of the field-dependence of the chemical potential. This is a dangerous omission in many papers. However, in most cases it is not relevant as it proved to be here. For clarity, this entire discussion should be moved to the supplement and merely the result stated.

- The discussion about the a-direction magnetostriction is very confusing. What did we now learn? The authors state that in their setup the angle fine-adjustment is difficult and they do not know exactly when the field is aligned with the axis. That is quite common of an experimental challenge. However, As they show, the magnetostriction depends very sensitively on the alignment, a few degrees off the c-direction the curves look very different. Do the authors have an explanation for this somewhat unexpected behavior close to a symmetry direction? Furthermore, can they exclude that a similar alignment sensitivity exists for the c-direction experiments shown earlier?

Reply to the remarks of Reviewer #1

We highly appreciate a careful reading of our manuscript by the Reviewer. We are glad that the Reviewer #1 finds our work interesting and worth further consideration. Taking advantage of her/his in-depth insight into the manuscript, we have revised the main text as well as substantially extended the supplementary information to brighten the essence of our work. Below are replies to all the questions and remarks raised by the Reviewer #1:

1. *The authors argue that the lowest Landau level may rise above the Fermi energy when the δ is less than $1/2$ for trivial electrons. Do you mean the Fermi energy is fixed in the system? It is not physical since there are electrons without the magnetic field if all the electrons belong to a single band. Where do these electrons go in this extremely large field? Actually, you argue below Eq. (1) "formula (1) predicts that the magnetostriction vanishes since $n_i(B) = n_i(0)$ due to the conservation of the carriers". The discussion for $\delta < 1/2$ is inconsistent with this argument.*

Within the rigid band approximation, the magnetostriction of a crystal with a single Fermi pocket vanishes. Hence, this quantity is interesting only for crystals with multiple nonequivalent pockets, and Weyl semimetals fall into this class. In the beginning of the manuscript (*Distinction between the Weyl and trivial electrons*), we really consider the case of constant chemical potential since as shown in *Analysis of the magnetostriction for TaAs*, this case well describes the realistic situation when the *total* charge-carrier density is conserved. We now distinctly say this on page 2.

2. *If $\delta > 1/2$, at least one level remains occupied at fixed Fermi energy case, and in the ultra-quantum limit, the magnetostriction is proportional to $B^{3/2}$. Further, it is linearly dependent on the magnetic field for Weyl electrons. However, it is hard to distinguish from each other. We can see from the inset of Fig. 1 that there is no obvious difference between the Weyl and trivial electrons just from two line shapes in the quantum limit. Hence, I think if you fit data of the magnetostriction along [001] direction by using $B^{3/2}$ but not B , you may obtain almost the same value of the coefficient b . I know if you fit the data with the trivial model, the mismatch of the phase in the oscillation part before the quantum limit will occur, which could exclude the trivial one. Then one could find out the Weyl electrons directly from the phase shift of the oscillation and it seems there is no need to determine the relativistic electrons from the data fit in the quantum limit.*

In a finite interval of B , the shape of the line $B^{3/2}$ indeed looks like the shape of the linear-in- B curve, but the slopes of this lines (i.e., the quantities λ) are different. We now discuss this issue in more detail in Supplementary Note 3 [after Eqs. (21)]. It is also indicated after formulas (7) in the revised main text that the parameters found from the simple analysis of the experimental data in the ultra-quantum limit are then verified by the calculation of the magnetostriction at *all* B , for which the experimental data can be obtained. (This comparison includes the check of the phase of the oscillation, see the end of *Results*). Generally speaking, if the $B^{3/2}$ dependence for the trivial electrons in the ultra-quantum region is interpreted as a manifestation of the Weyl electrons and is approximated by a linear function, its slope will lead to an incorrect value of the parameter b . This incorrect value will not permit one to reproduce the experimental data in the whole interval of the magnetic fields. The case of TaAs for $B \parallel [001]$ is discussed in point 3 below.

3. I find the authors also plotted the derived coefficient λ versus the magnetic field in the bottom of Fig. 2(a). The behavior of λ in the quantum limit may help one to distinguish the Weyl electron from the trivial one. But I am not sure whether it is true. Further, it is linear in the quantum limit. Does the intercept of the straight line on the vertical coordinate is the coefficient b ? If the answer is true it should be compared with the one from the fit of magnetostriction above 8 T?

We fully agree with the Reviewer #1 that the quantity λ does be useful in estimating the values of the parameters. For the longitudinal magnetostriction of TaAs along the [001] direction measured up to 16 T, i.e., in the weak-field range for the trivial charge carriers, it is correct that the intercept of the straight line corresponds to the parameter b (cf. the dashed line with $b = 2.28e-7$ in the revised version of Fig. 2a). The intercepts of admissible straight lines approximating our experimental data lay in a certain interval $(1.97-2.55)e-7$, and our fit of the magnetostriction above 8 T gives the value of b being in good agreement with the upper limit of this interval. In addition, we note that the slope of the dashed straight line in Fig. 2(a) corresponds to the parameter $2*c_h$ and amounts to $2.95e-8$. This value is in good agreement with the fit (see Table I), and thus provides further support for our interpretation.

4. The authors mentioned that for magnetization not the magnetostriction, a large electron group (or even a filled energy band) will give a large contribution to the magnetization proportional to B for Weyl electrons in the ultra-quantum regime, which may be indistinguishable from the term $B \ln(B)$ produced by the Weyl electrons. However, both the magnetostriction and the magnetization are thermodynamic probe. Why does the large electron group not affect the magnetostriction?

The magnetization and the magnetostriction result from different parts of the free energy of the crystal. The magnetization is due to the change of the electron energy in the magnetic field, whereas the magnetostriction characterizes the change of the interaction energy between the electrons and the elastic degrees of freedom in the crystal. The magnetic field changes the energy of a band filled by electrons and of a large group of the trivial electrons (this change increases with increasing size of the group), and these facts explain the above-mentioned feature of the magnetization. On the other hand, the change of the interaction energy within the rigid-band approximation is determined by the change of the charge-carrier density in the magnetic field. This change is absent for the filled bands, and it *decreases* with increasing size of the group. We discuss this issue in more detail in *Advantages of the magnetostriction* (main text) and new Supplementary Note 4.

5. I find the authors emphasized that the magnetostriction relates to n_i , the density of the carriers, but not the density of the states like the resistivity. What is the advantage of it?

The density of states decreases with decreasing size of the charge-carrier group, and so a large group of trivial electrons masks small pockets of the Weyl electrons in quantities that are proportional to the density of states. On the other hand, the change of the charge-carrier density in the magnetic field, which determines the magnetostriction, increases with decreasing the density. Now, this aspect of dilatometric experiments is wider discussed in *Advantages of the magnetostriction*.

6. There are two kinds of Weyl nodes with different cross-sectional areas contributing to the magnetostriction. Especially, the shape of the W1 pocket is banana-like. How does this shape affect your results?

The banana-like shape of the W1 pockets, which was found in the band-structure calculation, indicates that these pockets, strictly speaking, are not ellipsoids since their longest axis is slightly curved (Fig.3c in [20]). However, this curvature is small, and we neglect it. To consider the effect of this curvature, one should take into account relatively small higher-order terms (in quasimomentum) in the k-p Hamiltonian of the Weyl electrons. This is very difficult mathematical problem, but these terms will give only small corrections to our results, and so the effect of these terms is beyond the scope of our work.

7. The authors fit the measured magnetostriction along [001] in Fig. 3 by a constant chemical potential or Fermi energy. Further, they argued that the B-dependence of the chemical potential could be neglected. It could be clearer if they also plot the calculated magnetostriction for the fixed electron density in Fig. 3. Actually, the Fermi energy alters a lot in this ultra-quantum limit shown in Fig. 4. Hence, the direct comparison may be more compelling.

Thank you for this recommendation, which indeed enhances readability of our theoretical results. In Figs. 3 and 5, we now compare the experimental data on the magnetostriction with the theoretical curves calculated both assuming the constancy of the chemical potential ζ and taking into account the dependence $\zeta(B)$.

8. The magnetostriction measured along the a axis shows strong dependence on the angle between the magnetic field and the a axis. Particularly, it is positive at 81 degree, negative at 86 degree, and positive again at 90 degree. Why does the sign of the magnetostriction rely on this angle so strongly? Further, this magnetostriction measured along the a axis could not be fitted by just varying the cross sectional area. A proper fit could be realized by using a completely different set of parameters. It is not convincing.

This angular dependence of the a-axis magnetostriction is a complex and yet intriguing aspect of the field-induced length change in TaAs. It points at the strong anisotropy of the thermodynamic properties in the relatively simple tetragonal structure of the TaAs family, and thus indirectly emphasizes the significance of the longitudinal magnetostriction along the [001] direction. Nevertheless, we agree that this part of our manuscript has not been clear and convincing enough. Therefore, we have rewritten the manuscript, and a possible reason of the strong angular dependence of the a-axis magnetostriction is now discussed in Supplementary Note 8. We also explain in the revised main text that two sets of the parameters are admissible for describing the c-axis magnetostriction of TaAs, but one of these sets does not permit us to describe the a-axis magnetostriction.

Reply to the remarks of Reviewer #2

We greatly value that the Reviewer #2 personally likes our work and subject very much, but we are disappointed that she/he does not see this as a sufficient increment over the status-quo of the field to warrant a publication in Nature Communications. We have substantially revised the manuscript in order to highlight novelty of our magnetostriction study and to take into account the other remarks of Reviewer #2. We believe that she/he will now recognize this multidisciplinary high-rank journal as an appropriate platform to present our research.

We now provide point-by-point response to the remarks of Reviewer #2.

Let me lay out my rationale. The anomalous behavior of Landau levels formed from Weyl fermions is well established, also by the authors works, and has been observed theoretically in

the magnetization and the magnetic torque. There, the same mechanism leads to an anomaly that involves a sign change and very sharp features as the sample enters the quantum limit. The novelty of this work lies in its observation in magnetostriction, yet this is merely choosing a different channel to probe the free energy anomaly. Certainly the paper does not show any new physics about TaAs or ultra-quantum-limit physics, it shows that the magnetostriction is compatible with published results. This is wonderful, yet by itself too much of a nuance for a Nature Communication paper.’’

We respectfully disagree with the statement that the physics of the magnetization and the magnetostriction is identical. The physics of the magnetostriction differs from the physics of the magnetization, because these quantities probe different parts of the free energy. The magnetization results from the change of the electron energy in the magnetic field, whereas the magnetostriction is due to the energy of the interaction between the electron and elastic degrees of freedom in a crystal. This distinction between the energies leads to the different physics for the magnetostriction and magnetization, and we discuss this difference in detail in the revised manuscript (see *Advantages of the magnetostriction*) and new Supplementary Note 4, emphasizing that they give complementary information on parameters of Weyl semimetals.

Having the above point in mind, an importance of our magnetostriction study is illustrated on a prototype topological Weyl semimetal TaAs. For this material, the magnetic-torque anomaly [18] in the ultra-quantum limit (QL) is absent, and thus the evidence for relativistic quasiparticles comes from non-saturating linear magnetization (measured using angle-dependent torque magnetometry) much above the fields at which *both* relativistic and conventional quasiparticles enter their QLs [19]. However, for the longitudinal magnetostriction along the [001] (degeneracy of the Fermi-surface pockets is conserved), the thermodynamic evidence for relativistic quasiparticles has been found when *only* relativistic fermions are in the QL. In this arrangement, applied magnetic fields serve as the low-field limit for the trivial holes with the 19-T fundamental frequency.

The novelty of our study also lies in the fact that we do not apply simplified models of the Weyl spectra when a tilt inherent in these spectra is disregarded, when several Weyl pockets are replaced by one, and when the conservation of the particles is considered only for a single pocket, and not for all the charge-carrier groups together. In other words, we establish the approach that can be used for detecting the magnetostrictive response of relativistic quasiparticles in real non-centrosymmetric Weyl semimetals. We also demonstrate how the parameters Λ_i , which do not appear in formulas for the magnetization, can be found from the magnetostriction (Table II).

Besides, we communicate the first-ever results for magnetostriction in a topological semimetal, indicative of large anisotropy effect that experiences immense changes with minute deviations of the applied field direction. The magnetostrictive stress may be relevant for future high-field Weyltronic devices, since strained thin-films can have distinctly different properties from an unstrained material. We also suppose that other groups will benefit from our research, since a number of high-resolution dilatometry cells are already used on a PPMS and standard dc magnets can be sufficient to detect relativistic fermions by magnetostriction.

‘‘Before going into the details, I want to emphasize a main point of critique. I find the description of previous works, including some of the authors, to be brushed away too rigorously in the introduction. It is stated that magnetic susceptibility is a useful probe, yet then these signatures are supposedly ‘‘subtle and indirect’’. Accidentally convolving the magnetic anomalies with the issues of negative LMR here is dishonest and does not do the paper service. The later discussion on the comparison between forms of magnetization and magnetostriction appears very much as a ‘‘sales argument’’ rather than an open discussion. The goal of this

comparison is to argue that magnetostriction is superior as a technique, as is already evident from the title. I disagree with this spirit, both are interesting probes in their own right that elucidate different aspects of a physical object. As the physics is identical, the emphasis is on practical aspects such as field alignment and background subtraction. ...

I would suggest to rewrite this paper to highlight what new physics we can learn. Magnetostriction does not need to be “superior” to magnetization; maybe combining the techniques truly unlocks new insights.’’

We regret to note that the Referee got such an impression. This is not our intention to contrast magnetization and magnetostriction in such a manner. Therefore, we have modified the title, the abstract, the introduction, and the comparison of these quantities in *Advantages of the magnetostriction* (main text). In the revised version, our comparison indicates useful features of both the magnetostriction and the magnetization. We also emphasize that since the magnetization and magnetostriction probes different parts of the free energy, they give complementary information on parameters of the Weyl semimetals (see Table II for the parameters λ_i , which cannot be found from the magnetization).

Below we answer the specific remarks and recommendations of the Reviewer #2:

- *From a practical point of view, magnetization measurements are much simpler and prevalent compared to magnetostriction. Even so, the authors conclude that the fitting parameters do not yield a self-consistent picture once the magnetostriction along the a-direction is taken into account.*

We agree that from a practical point of view, the torque measurements, which give the transverse component of the magnetization, are simpler than the experiments with the magnetostriction, but this component of the magnetization vanishes for the magnetic fields directed along symmetry axes. On the other hand, for the magnetic fields tilted from the symmetry axes, the *theoretical analysis* of the magnetization (and of the magnetostriction) is much more complicated than in the symmetric cases if one take into account a realistic Fermi-surface topology with a large number of the Fermi-surface pockets. For this reason, the symmetric situations, directly probed by the magnetostriction, are more convenient for the *initial* analysis of the Weyl semimetals. Furthermore, we would like to add that measurements of the longitudinal magnetization in high magnetic fields (i.e., in fields higher than an upper limit for SQUID magnetometry of about 7 T) give much less accurate results.

In the revised manuscript, we also explain that only one of the two possible sets of the parameters does not yield a self-consistent picture if the a-axis magnetostriction is taken into account. For the other set, the self-consistent description is possible, but in this case, the dispersion of the holes in TaAs can hardly be described by a simple parabolic law.

- *The rigid band approximation is well justified in metals, yet I wonder if it is as simple in the case of Weyl fermions. For once, the involved energy scales are meV, `on the order of 10^{-3} of the total bandwidth. Is it obvious that on these energy scales rigid band is always justified? These nodes may also be moved along equal energy contours. It may be, but a bit of justification would be helpful.*

We now discuss the rigid-band approximation in Supplementary Note 1.

- *The authors stress about eq (6) that it allows to obtain the FS cross-sections F_i , which are usually measured by quantum oscillation frequencies, even when there are no oscillations. This appears a bit misleading. It essentially calculates the area from the knowledge of the carrier density and computes the filling required to obtain it. Similarly, eq (5) would in principle allow to obtain the F_i without oscillations, yet it similarly needs stringent assumptions on the dispersion. Lastly, these are band-dependent contributions which are not directly measurable by experiment, which always gets the sum. This sentence goes too far in my opinion and should be removed.*

We agree that this sentence could be misleading. We have changed the appropriate part of the text, and this sentence is now absent.

- *I find it strange that $W1$ is observed so vividly yet $W2$ is completely absent. This might be due to that difference in scattering rates as the authors suggest (yet the a -direction magnetorestriction yields γ values that are incompatible with this and would suggest stronger oscillations of $W2$). It would however require strongly k -selective scattering mechanisms.*

So far, there is no clear explanation of the fact that the frequency F_{W2} is unobservable for B parallel to the c axis. Arnold et al. [30] have presented the most conclusive investigations of the Fermi surface topology of TaAs by means of angle-dependent measurements of quantum oscillations (QOs) in magnetization, magnetic torque, and magnetoresistance, and they did NOT resolve QOs originating from the $W2$ electron pockets for B parallel to the $[001]$ direction. (Only for other field directions, the very weak dHvA signals due to the $W2$ pockets were detected in the narrow magnetic window from 1.4 to 3 T; cf. Fig. S7 in [30]). It should be also noted that the $W2$ QOs for B along the c axis have not been resolved in any other experiments, including magnetic torque [19], thermal conductivity, thermopower, and magnetization [13] as well as recently performed measurements of field-angle dependence of sound velocity [31].

Our notations of γ_{Wi} seem to be unclear in the old version of the manuscript, and so they led to a misunderstanding. In the old variant, the Dingle temperature for $W2$ electrons was normalized to the Fermi energy measured from the $W2$ points when we considered the case of the constant chemical potential. The appropriate dimensionless ratio was equal to 0.1 (page 5). However, when we considered the B dependence of the chemical potential, the parameter γ_{W2} was defined as this Dingle temperature normalized to the Fermi energy measured from the $W1$ point, and this γ was equal to 0.04 (see Table I in the old variant of the manuscript). In reality, the same Dingle temperature was used in both the cases ($T_{D,W2} \sim 3.6$ K), and the difference in the values of γ was due to the normalized factors which differed in 2.5 times. To avoid this confusion, we use one and the same normalization throughout the revised manuscript. Note that for the a -axis magnetostriction, which was measured for another sample, we have to increase the Dingle temperatures for the $W1$ and $W2$ electrons in order to suppress the oscillations.

- *Conveniently, there are 2 Weyl pockets which allows to solve the polynomial for a_{W1} and a_{W2} . While implicitly stated in the discussions about field angles, it would be beneficial to state this clearly when discussing the calculation.*

We now clearly state this in page 5 after Eqs. (10). Note that for TaAs, this convenience is due to the symmetry of the situation that we consider (B is along the c axis).

• I highly appreciate the robust treatment of the field-dependence of the chemical potential. This is a dangerous omission in many papers. However, in most cases it is not relevant as it proved to be here. For clarity, this entire discussion should be moved to the supplement and merely the result stated.

We are glad to note that the Referee appreciates our concern to figure out the field-dependence of the chemical potential. While we indeed do not observe a considerable effect of this dependence on the magnetostriction of TaAs up to 16 T, our work demonstrates how one can find out the role of this dependence in other materials of the TaAs family. In the revised manuscript, we moved the discussion of the field dependence of the chemical potential to Supplementary Note 6, and in the main text (in page 7), only the results of our analysis are presented.

• The discussion about the a-direction magnetostriction is very confusing. What did we now learn? The authors state that in their setup the angle fine-adjustment is difficult and they do not know exactly when the field is aligned with the axis. That is quite common of an experimental challenge. However, as they show, the magnetostriction depends very sensitively on the alignment, a few degrees off the c-direction the curves look very different. Do the authors have an explanation for this somewhat unexpected behavior close to a symmetry direction? Furthermore, can they exclude that a similar alignment sensitivity exists for the c-direction experiments shown earlier?

The high sensitivity of the a-axis magnetostriction to the direction of B near the c axis was also unexpected for us since we do not observe a similar sensitivity for the c-axis magnetostriction. We call attention of the physical community to this unusual result, and a possible explanation of this high sensitivity is now presented in Supplementary Note 8.

REVIEWER COMMENTS

Reviewer #1 (Remarks to the Author):

The authors have addressed most of my comments properly. On the magnetostriction, I think that it is new and interesting, different from the magnetization or torque measurements. The explanation on the sensitive angle dependence of the magnetostriction is reasonable to me. I recommend the publication of the manuscript in Nature Communications.

Reviewer #2 (Remarks to the Author):

This is my second reading of the paper now called "Detection of relativistic fermions in Weyl semimetal TaAs by magnetostriction measurements" by T. Cichorek.

During the rebuttal period, the world has changed. I hope for the safety of all Ukrainians, and deeply wish that our colleagues Mikitik and Sharlai are safe.

I appreciate the change in the paper and tone, as well as the extensive and thoughtful response to my and referee 1's questions. In particular, the role of magnetostriction as a distinct and complementary quantity became clearer. I still think that the extent of the discussion of TaAs with the two possible sets of parameters is a bit too lengthy for a Nature Communications article, especially as it remains inconclusive about the physics of TaAs. I am inclined to recommend the paper for publication, yet there are a few minor points I would recommend the authors to consider:

- "That is why the magnetostriction is small (though still detectable [21])" (P2). It is confusing that eq.1 predicts the absence of magnetostriction for single-band metals, yet here it states that it remains small yet detectable. Which assumptions in Supplement 1 and the derivation of (1) are violated in real materials that it is non-zero? It is a quite general thermodynamic argument? Maybe best remove this here, it's not that central if it truly vanishes for a single band material as all Weyl systems of interest are multi-band.

- In what capacity is [21] cited for this statement? As far as I understand, [21] is a torque study on NbP in the quantum limit. It does not seem right here to reference non-vanishing magnetostriction in single-band materials.

- "Hence, in measurements of these quantities, experimental signatures for small Fermi-surface pockets are extensively masked by the contribution of a large pocket when it exists in the material." (P3). I disagree with this new statement that quantities like magnetization or resistivity, which depend on the DOS(E_F), show therefore less pronounced quantum oscillations of small pockets (which are naturally enhanced in magnetostriction). There are numerous counterexamples, for example LaRhIn5 which the authors are famous for. In this heavily multi-band metal, the magnetization and the resistivity is dominated by large oscillations of a tiny $F \sim 7$ T pocket whereby the large $F > 1000$ T pockets are faint and barely detectable (R.G. Goodrich et al., PRL 89, 26401 (2002); C. Guo et al., Nat. Comm 12:6213 (2021)). I do not think this discussion about pocket size and detectability is justified and should be removed / adopted.

- How was the FFT in Fig. 2 computed, and was a single dampening taken into account? At such low frequencies and few oscillations, the field-dependent envelope will leap into the frequency spectrum, likely giving rise to this broad frequency smearing in the inset.

- "As in all previous experiments with TaAs [13, 19, 30, 31], the oscillations with the frequency ω do not manifest themselves in our data on the magnetostriction" (p5). This is correct. It would be nice to early on add what could be an origin for this and / or how the authors achieve this in the theory. Obviously they simply add scattering onto this pocket, but why such sharply k -selective scattering should exist in TaAs is a mystery. Somehow, the low-frequency scenario of the authors provides a more natural explanation, as even with the same scattering coefficient naturally no oscillations could be observed for such a low frequency.

- Lastly, and this is more of a comment, I wonder if the discussion on the magnetic torque vanishing for fields along the principle crystal directions could be mitigated by higher-order derivative techniques. For example, the field-derivative of the torque, $d^2F/d\theta^2 = d^2 \text{torque} / d\theta^2$, also addresses this issue in a purely magnetic measurement, as has been shown in NbP (K.A. Modic et al., Nat. Comm:3975 (2018)).

I feel addressing these points could strengthen the paper, and am happy to recommend it for publication.

Reply to the remarks of Reviewer #2

We highly appreciate the double reading of our manuscript by the Reviewer #2. In particular, a few additional comments to strengthen the quality of the manuscript. Most of them are incorporated in the revised version (red color) as indicated below:

1) “That is why the magnetostriction is small (though still detectable [21])” (P2). It is confusing that eq.1 predicts the absence of magnetostriction for single-band metals, yet here it states that it remains small yet detectable. Which assumptions in Supplement 1 and the derivation of (1) are violated in real materials that it is non-zero? It is a quite general thermodynamic argument? Maybe best remove this here, its not that central if it truly vanishes for a single band material as all Weyl systems of interest are multi-band.

If the assumption of the rigid bands were exact, the magnetostriction of a single-band material would be zero. However, this assumption is only a useful approximation (see the end of Supplementary Note 1), which was used in deriving Eq.(1). The very small nonzero value of the magnetostriction in real single-band metals is due to a violation of this approximation and characterizes its accuracy. This accuracy was experimentally demonstrated in Ref. [21], in which the magnetostriction of bismuth containing electron and hole pockets was measured. When the sample of Bi was doped by Sn, then the electron pockets completely disappeared and only single hole pocket remained. The magnetostriction of the doped sample sharply decreased as compared to pure Bi, giving rise to essentially flat behavior near the zero value (see Fig.5 in [21]).

We agree that our phrase in p.2 was not clear, and so we have rewritten it in the revised manuscript.

2) In what capacity is [21] cited for this statement? As far as I understand, [21] is a torque study on NbP in the quantum limit. It does not seem right here to reference non-vanishing magnetostriction in single-band materials.

The Reviewer seems to confuse Refs. [20] and [21]. In Ref. [21] the magnetostriction of Bi was considered (see above).

3) “Hence, in measurements of these quantities, experimental signatures for small Fermi-surface pockets are extensively masked by the contribution of a large pocket when it exists in the material.” (P3). I disagree with this new statement that quantities like magnetization or resistivity, which depend on the $DOS(E_F)$, show therefore less pronounced quantum oscillations of small pockets (which are naturally enhanced in magnetostriction). There is numerous counterexamples, for example $LaRhIn_5$ which the authors are famous for. In this heavily multi-band metal, the magnetization and the resistivity is dominated by large oscillations of a tiny $F \sim 7T$ pocket whereby the large $F > 1000T$ pockets are faint and barely detectable (R.G. Goodrich et al., PRL 89, 26401 (2002); C. Guo et al., Nat. Comm 12:6213 (2021)). I do not think this discussion about pocket size and detectability is justified and should be removed / adopted.

Before the contentious phrase, we refer to quantities which are proportional to the density of the charge-carrier states at $B=0$, i.e., we imply the smooth (non-oscillating with B) part of the electrical conductivity. To avoid this misunderstanding we have more accurately rewritten the preceding sentences (page 3, right column). We would like also to note that we do not mention the magnetization here since its orbital part is not determined by the $DOS(E_F)$, and the magnetization increases with decreasing size of a pocket surrounding a Weyl (Dirac) point or a band-contact line. For this reason, the magnetization of $LaRhIn_5$ is dominated by the tiny pocket.

4) How was the FFT in Fig. 2 computed, and was a dingle dampening taken into account? At such low frequencies and few oscillations, the field-dependent envelope will leap into the frequency spectrum, likely giving rise to this broad frequency smearing in the inset.

A smooth background $\propto B^2$ was subtracted from the low field magnetostriction data measured at $T = 25$ mK. The inset in Fig. 2 (a) shows the corresponding Fourier transformation of the background subtracted magnetostriction signal over a magnetic range of 1 to 8 T. We also find that a decrease in the magnetic field range where the FFT is carried out practically does not shift the position of the

maximum at 7.2 T. In the revised version, the caption to the inset of Fig. 2(a) provides more details about the FFT.

We agree with the comment about the subtle points of the FFT, but we would like to emphasize that our formulas for the magnetostriction describe its oscillating and smooth parts together, and we find the dominant $F_{\alpha} = 7.2$ T frequency from the fit of the theoretical curve (for which the Dingle damping was taken into account) to the experimental data. The FFT data shown in the inset of Fig. 2(a) only demonstrate that there is no contradiction of our results obtained from the fit with the traditional approach.

5) "As in all previous experiments with TaAs [13, 19, 30, 31], the oscillations with the frequency F_{W2} do not manifest themselves in our data on the magnetostriction" (p.5). This is correct. It would be nice to early on add what could be an origin for this and / or how the authors achieve this in the theory. Obviously they simply add scattering onto this pocket, but why such sharply k-selective scattering should exist in TaAs is a mystery. Somehow, the low-frequency scenario of the authors provides a more natural explanation, as even with the same scattering coefficient naturally no oscillations could be observed for such a low frequency.

The Reviewer says about the k-selective scattering since our $\gamma_{W1}=0.025$ and $\gamma_{W2}=0.1$ are noticeably different. However, these γ correspond to $T_{D,W1}=2.2$ K and $T_{D,W2}=3.6$ K (p.6), the difference of which is not so large. Nevertheless, these Dingle temperatures do lead to different scattering times $\tau_i \sim \hbar/T_{D,Wi}$. However, according to the calculation of Ref.[24] now cited in Supplementary Note 7, the velocities V_i of the W1 and W2 electrons differ by a factor of about two, and therefore these W1 and W2 electrons have approximately the same mean-free path ($\tau_i V_i$) as this occurs for W1 electrons and holes in TaAs [30]. Thus, the difference in γ does not necessary mean that the k-selective scattering takes place. We appreciate that the Reviewer #2 found interesting a possible scenario that the absence of the oscillations for W2 electrons can be explained by the small value of F_{W2} . Taking into account all the points raised by the Reviewer, we now discuss them in revised Supplementary Note 7.

6) Lastly, and this is more of a comment, I wonder if the discussion on the magnetic torque vanishing for fields along the principle crystal directions could be mitigated by higher-order derivative techniques. For example, the field-derivative of the torque, $d^2F/d\theta^2 = d \text{ torque} / d\theta$, also addresses this issue in a purely magnetic measurement, as has been shown in NbP (K.A. Modic et al., Nat. Comm: 3975 (2018)).

We thank the Reviewer #2 who called our attention to the interesting paper of Modic et al. In that paper, a new measurable quantity (magnetotropic coefficient), which does not vanish for the magnetic fields along the principle crystal directions, was indicated. At the end of "Advantages of the magnetostriction", we now tell about this quantity and cite the paper by Modic et al. (new Ref. [26]).

PS Thank you for wishes to GPM and YVS. I know it was important for both of them. They are safe, although GPM decided to stay in Kharkiv. YVS with his wife leaved Ukraine on March 8th and spend a few weeks with my family. Recently, he obtained a reasonable financial support from the Polish National Science Centre and now is working with me in Wroclaw. TC